# Treatment-Related Adverse Events of Combination EGFR Tyrosine Kinase Inhibitor and Immune Checkpoint Inhibitor in EGFR-Mutant Advanced Non-Small Cell Lung Cancer: A Systematic Review and Meta-Analysis

**DOI:** 10.3390/cancers14092157

**Published:** 2022-04-26

**Authors:** Daisy Wai-Ka Chan, Horace Cheuk-Wai Choi, Victor Ho-Fun Lee

**Affiliations:** 1LKS Faculty of Medicine, The University of Hong Kong, Hong Kong, China; daisycwk@connect.hku.hk; 2School of Public Health, LKS Faculty of Medicine, The University of Hong Kong, Hong Kong, China; hcchoi@hku.hk; 3Department of Clinical Oncology, School of Clinical Medicine, LKS Faculty of Medicine, The University of Hong Kong, Hong Kong, China; 4Clinical Oncology Center, The University of Hong Kong-Shenzhen Hospital, Shenzhen 518053, China

**Keywords:** NSCLC, EGFR mutation, tyrosine kinase inhibitors, immune checkpoint inhibitors, meta-analysis

## Abstract

**Simple Summary:**

The use of combination epidermal growth factor tyrosine kinase inhibitor (EGFR-TKI) and immune checkpoint inhibitor (ICI) in EGFR-mutant, advanced non-small cell lung cancer (NSCLC) has raised concerns over the risk of overlapping toxicities. Although a higher proportion of interstitial lung diseases was reported with the combination of osimertinib and durvalumab, the current evidence on the treatment-related adverse events (trAEs) of EGFR-TKI and ICI remains limited to pneumonitis and the use of osimertinib. This systematic review and meta-analysis investigates whether combination EGFR-TKI and ICI increases the incidence of overall and organ-specific trAEs compared to TKI monotherapy. A higher proportion of high-grade organ-specific trAEs was observed in combination TKI and ICI including skin, gastrointestinal adverse events, and interstitial lung diseases. Further prospective studies are warranted to determine the optimal drug of choice of ICI and the best timing of initiation of ICI after failure to prior EGFR-TKI.

**Abstract:**

(1) Background: We performed a meta-analysis to examine whether combined epidermal growth factor tyrosine kinase inhibitor (EGFR-TKI) and immune checkpoint inhibitor (ICI) increases treatment-related adverse events (trAEs) in advanced non-small cell lung cancer (NSCLC). (2) Methods: Articles from MEDLINE, EMBASE, and Cochrane databases were searched. Proportions and odds ratios (ORs) of the pooled incidence of overall and organ-specific trAEs in combination EGFR-TKI and ICI were compared to TKI monotherapy. (3) Results: Eight studies fulfilled our selection criteria. Any-grade organ-specific trAEs were more common in combination EGFR-TKI and ICI than TKI monotherapy (skin: OR = 1.19, *p* = 0.012; gastrointestinal tract: OR = 1.04, *p* = 0.790; ILD: OR = 1.28, *p* = 0.001). Grade ≥ 3 trAEs were also more frequent in combination treatment (skin: OR = 1.13, *p* = 0.082; gastrointestinal tract: OR = 1.13, *p* = 0.076; ILD: OR = 1.16, *p* = 0.003). (4) Conclusions: A higher proportion of grade ≥3 skin and gastrointestinal trAEs and ILDs was observed in combination TKI and ICI compared to TKI alone. Caution has to be taken when interpreting the results owing to the small number of studies included in this meta-analysis.

## 1. Introduction

Sensitizing somatic mutations of the epidermal growth factor receptor (EGFR) gene were found in about 10% of Caucasian and 50% of east Asian patients with non-small cell lung cancer (NSCLC) [1,2,3]. EGFR tyrosine kinase inhibitors (TKIs) have been the standard first-line treatment for EGFR-mutated advanced/metastatic NSCLC. Despite the evolving generations of TKIs, e.g., dacomitinib and osimertinib, patients usually developed acquired drug resistance after a median progression-free survival (PFS) of 9 to 19 months [4,5,6,7,8,9,10,11,12,13,14,15,16]. While chemotherapy with or without other systemic anti-cancer treatments was an acceptable second-line therapy following progression against TKI, especially in patients who had already failed osimertinib or do not harbor exon 20 T790M mutation [17,18], maintaining the original TKI with the addition of another anti-cancer systemic treatment, e.g., chemotherapy, antiangiogenic agents, or immune checkpoint inhibitors (ICI), were also possible appealing treatment options, particularly in patients whose tumor still harbored the pre-existing EGFR mutations confirmed by tumor or liquid re-biopsy [19,20,21,22,23,24,25,26,27,28,29,30,31,32,33].

Recently, there have been a number of clinical trials and retrospective studies on using ICI targeting programmed death 1 (PD-1) and its ligand (PD-L1), either alone or in combination with TKI in patients who had EGFR-mutated NSCLC after failure to prior TKIs [34,35,36,37,38,39]. However, either combinational or sequential use of EGFR-TKI and ICI raised concerns over the risks of overlapping toxicities. In both the TATTON and CAURAL studies, a high incidence of interstitial lung disease (ILD) was reported with the combination of osimertinib and durvalumab, leading to premature closure of the studies [35]. The results concurred with the analysis, based on the US Food and Drug Administration Adverse Event Reporting System (FAERS) database, that reported a higher proportion of interstitial pneumonitis for nivolumab in combination with EGFR-TKIs versus nivolumab alone [36]. A recent study suggested that severe immune-related adverse events may be associated with the sequential administration of anti-PD-L1 inhibitors followed by osimertinib [37]. Other studies also reported a higher incidence of either severe hepatotoxicity or interstitial lung diseases on using osimertinib immediately (within 180 days) after nivolumab [38,39]. However, this association appeared to be specific for osimertinib, as first or second-generation TKI administered immediately after ICI did not demonstrate excessive toxicity and ILD.

Since osimertinib has a much shorter half-life (T_1/2_ 55 h) [40] than anti-PD-(L)1 blockers such as nivolumab (T_1/2_ 20 weeks) [41], the prolonged receptor occupancy of anti-PD-L1 antibodies may increase the frequency of toxicities when administering osimertinib immediately after nivolumab. Nevertheless, the underlying mechanisms to explain the overlapping toxicity of these two distinct therapies are still undeciphered. EGFR-TKIs demonstrated enhanced T-cell-mediated killing of tumor cells by increasing both basal and IFN gamma induced MHC class-I presentation [42]. Thus, EGFR-TKIs may have underappreciated immunomodulatory effects that pose a risk of cross-reactions with normal tissues. In addition, a recent mouse model study described higher levels of proinflammatory cytokines (IFN-γ, IL-2, IL-5, TNF-α, and IL-12p70) in osimertinib and anti-PD-L1 combination than osimertinib monotherapy, indicating that the combined therapy promoted the release of inflammatory factors and exacerbated damage to normal lung tissues [43].

Despite the existing evidence that suggested increased risks of ILD in combination osimertinib and anti-PD-(L)1 inhibitors, most of these publications were either retrospective studies conducted in a single institution or early-phase clinical trials with small sample size. Few studies have investigated the frequency of other adverse events in other combinations of EGFR-TKIs plus ICI. Therefore, the objective of this study is to investigate the incidence of treatment-related adverse events (trAEs) of combination EGFR-TKI and ICIs in advanced EGFR-mutant NSCLC patients who have previously received TKI.

## 2. Materials and Methods

### 2.1. Search Strategy

This systematic review and meta-analysis (PROSPERO registration number: CRD42020184748) followed the Preferred Reporting Items for Systematic Reviews and Meta-analyses (PRISMA) guidelines [44]. PubMed-MEDLINE, Embase (Ovid), Scopus, Cochrane Library, and CINAHL were utilized to search for Phase I–III randomized controlled trials (RCTs), non-randomized controlled trials (non-RCTs), and retrospective studies without language restriction. The keywords were non-small cell lung cancer or non-small-cell lung cancer, EGFR, immunotherapy, and immune checkpoint inhibitors. The search date started from the inception of each database to 31 December 2021. Abstracts and presentations from conference proceedings including American Society of Clinical Oncology, World Conference on Lung Cancer, and European Society for Medical Oncology were also reviewed to identify unpublished studies. The clinicaltrials.gov database (https://www.clinicaltrials.gov, accessed on 31 December 2021) was also screened to identify clinical trials that had not been published.

### 2.2. Selection Criteria

RCTs, non-RCTs, single-arm studies, and retrospective studies that fulfilled the following criteria were included: (i) recruited patients aged 18 years and above with advanced NSCLC (stage IIIB–IV) who were previously treated with EGFR-TKI, (ii) evaluated the safety and tolerability of combination EGFR-TKI and ICI, (iii) administered EGFR-TKI and ICI regardless of sequence, either concurrently or sequentially, and (iv) reported incidence of trAEs that included any-grade and grade ≥ 3 events. Trials were excluded if they included patients who were TKI-naïve or used ICI in combination with chemotherapy or other regimens of ICI, were not available in full text, and were case reports.

### 2.3. Study Objectives

The primary objective was to evaluate the pooled incidence of overall and organ-specific including gastrointestinal, pulmonary, and dermatological trAEs of combination EGFR-TKI and ICI. The severity of trAEs was recorded as grade 1–5 based on version 3 or 4 of the Common Terminology Criteria for Adverse Events (CTCAE) of the National Cancer Institute (Bethesda, MD, USA). The secondary objective was to compare the pooled incidence of trAEs between combination therapy and TKI monotherapy. Previous literature suggested that osimertinib may carry a higher risk of trAEs and ILD [34,35,37,38,39]. As result, FLAURA [15] and AURA3 [45], the two registration trials on osimertinib as first-line treatment for patients with treatment-naive EGFR-mutant metastatic NSCLC and second-line treatment for patients with EGFR T790M mutation positive metastatic NSCLC after failure to prior TKI, as well as the osimertinib monotherapy arm of CAURAL study were selected as control for comparison of incidence of trAEs. The summary of osimertinib monotherapy studies is listed in Appendix A.

### 2.4. Data Extraction

Data extracted were recorded using Excel 2013 on a predesigned extraction list. For clinical trials, information extracted included: trial name, name of first author, year of publication, stage of NSCLC, EGFR-TKI treatment status, incidence of any-grade and grade 3 or above adverse events, and incidence of adverse events leading to drug discontinuation or death. For retrospective studies, patient characteristics, previous oncological treatment, trAE onset, current treatment details, and outcomes were recorded.

### 2.5. Assessment of Study Quality

The quality of RCTs was evaluated by the Cochrane Collaboration “risk of bias” tools [46], and that of non-RCTs was assessed by the Newcastle–Ottawa Scale (NOS) [47].

### 2.6. Outcome Measures and Statistical Analysis

The outcome measure was the proportion of trAE incidence. Freeman–Tukey double arcsine transformation was applied to transform proportions to continuous scale for meta-analysis. Odds ratios (ORs) were further generated to compare the pooled proportion of trAE incidence of combination therapy against TKI monotherapy. It has to be noted that one osimertinib monotherapy study was excluded from the pooled analysis of overall trAEs due to insufficient data available [48].

Statistical heterogeneity and inconsistency between the selected studies were assessed with the I^2^ statistics and τ^2^ statistics, respectively. If I^2^ > 50% or *p* value < 0.1 for the τ^2^ test, which indicated potential heterogeneity and inconsistency amongst the studies, a random effects model was used. Otherwise, the analysis was based on fixed effect models. We also performed sensitivity analysis on trAEs that had studies with at least moderate sample size ≥ 40 subjects reported in both monotherapy and combination groups.

Since patients’ individual data were not reported, the most common organ-specific disease was chosen to represent the incidence of the overall organ-specific trAE. For example, in Arm 1 of TATTON trial (osimertinib + duvalumab 3 mg/kg), vomiting (70%) was the most common gastrointestinal AE compared to nausea (30%) and diarrhea (30%). Therefore, the incidence of vomiting was chosen to represent the gastrointestinal AE of that arm. Statistical significance was defined as *p* value < 0.05 (two-sided). All data analysis was conducted using the statistical software R (version 3.6.3 (29 February 2020); R Foundation, Vienna, Austria) and the meta and metafor package.

## 3. Results

### 3.1. Search Strategy Results

A total of 4016 publications were identified by searching the literature bases, and two additional records were retrieved from manual searches (Figure 1). After abstract review and removal of duplicates, 40 full-text articles were selected for full-text screening. Three retrospective studies were excluded because they were case reports [49,50,51]. Three studies were excluded because patients did not previously fail TKI [52,53,54] and two studies did not include patients with EGFR mutations [33,55]. Three clinical trials were terminated [25,29,32] while 11 were not studies of combination ICI and TKI [19,20,22,56,57,58,59,60,61,62,63]. Ten ongoing trials from clinicaltrials.gov were excluded due to incomplete data (Appendix A) [24,26,27,28,64,65,66,67,68,69]. Therefore, the meta-analysis included three clinical trials (TATTON [34], CAURAL [35], and CheckMate 012 [70]) and five retrospective studies [36,37,38,39,71].

### 3.2. Study Characteristics of Clinical Trials

One RCT and two non-RCTs were selected, as summarized in Table 1. In total, 73 patients with stage IIIb/IV EGFR-mutant NSCLC were included. All were TKI-pretreated except for 1 patient in CheckMate 012 (NCT01454102) [70]. While CAURAL (NCT02454933) [35] was a phase III RCT, TATTON (NCT02143466) [34] and CheckMate 012 were phase I single-arm studies. Both TATTON and CAURAL evaluated the safety and tolerability of osimertinib plus durvalumab, while CheckMate 012 investigated the combination of erlotinib and nivolumab.

The median age of patients was 63 years (range 41–80) in CheckMate 012, 67 years (range 46–78) in arm 1 and 58 (range 44–73) in arm 2 of TATTON, and 65 years (range 41–80) in arm 1 and 56 (range 41–78) in arm 2 of CAURAL. The median duration of follow-up was 19.3 months (range 5.9–49.7) in CheckMate 012, but was not reported in TATTON and CAURAL.

All studies described the incidence of trAEs reported in ≥10% of patients graded according to CTCAE version 4. While CheckMate 012 reported organ-specific adverse events, other studies did not. As mentioned previously in the section “Data analysis”, if a study does not report organ-specific AE, the most common organ-specific disease was chosen to represent it. Only gastrointestinal and skin AE as well as interstitial lung diseases (ILD) had data reported by all studies. Hepatotoxicity was reported by CheckMate 012 and CAURAL but not in TATTON. Upper respiratory diseases were reported by CUARAL but not in CheckMate 012 and TATTON. Endocrinopathies were reported by CheckMate 012 but not in CAURAL and TATTON. Details on the incidence and nature of trAEs reported by each study can be found in Appendix A. The risk of bias of RCTs and the quality assessment of non-RCTs were also shown (Appendix A).

### 3.3. Study Characteristics of Retrospective Studies

All five retrospective studies reported ILD or pneumonitis in advanced, TKI pretreated NSCLC patients who have received sequential TKI and ICI therapy [36,37,38,39,71]. One study was based on the US Food and Drug Administration Adverse Event Reporting System (FAERS) database analysis [36], three were single-institution studies [37,38,39], and one was a multi-institutional study [71]. The summary of these studies was shown in Table 2.

### 3.4. Nature of trAEs Reported in Retrospective Studies

Four retrospective studies described the incidence of ILD in sequential or concurrent administration TKI and ICI [36,37,38,71]. Only one study reported severe immune-related adverse events other than ILD, including colitis and hepatitis [37]. Oshima et al. revealed a higher proportion of ILD for nivolumab in combination with EGFR TKI compared to treatment with either drug alone [36]. Uchida et al. found administration of first or second-generation TKI immediately after nivolumab did not increase the risk of ILD, but ILD was more frequent if osimertinib was used immediately after ICI [39]. Kotake et al. also highlighted an increased risk of ILD in the use of nivolumab immediately followed by osimertinib [38]. In this study, the median time to the onset of ILD from osimertinib initiation was 6 weeks (range 4–12) with a median interval of 2 weeks (range 1–4). In addition, PD-L1 blockade followed by osimertinib was associated with an increased risk of severe trAEs [37]. However, this association appeared to be specific to osimertinib, as no severe trAES occurred with the administration of other TKIs. In general, trAEs occurred at a median onset of 20 days (range 14–167 days) after osimertinib. In a recent retrospective study investigating the efficacy and safety of the use of ICI monotherapy after TKI, no ILD was reported. The ICI monotherapy included nivolumab, pembrolizumab, or atezolizumab and the median period between termination of TKI and initiation of ICI was 4.5 months (0.03–22.8) [71].

### 3.5. Overall Incidence of trAEs

The pooled incidence of overall trAEs in combination TKI and ICI and osimertinib monotherapy is shown in Table 3. The pooled proportion of overall trAEs was 100.0% (95% confidence interval (CI) 96.3–100.0) for any-grade and 30.0% (95% CI 12.0–51.6) for grade ≥ 3 toxicities in combination therapy, and 87.7% (95% CI 68.1–99.0) for any-grade and 13.8% (95% CI 0.1–40.4) for grade ≥ 3 toxicities in osimertinib monotherapy. In general, there was a higher risk of any-grade (odds ratios (OR) 1.27, 95% CI 0.75–1.66, *p* = 0.077) and grade ≥ 3 (OR 1.23, 95% CI 0.85–1.76, *p* = 0.270) overall trAEs in combination therapy than osimertinib monotherapy, though the significance was weaker in grade ≥ 3 events (Appendix A).

### 3.6. Incidence of Organ-Specific trAEs

Organ-specific trAEs (any-grade) in combination TKI and ICI were observed with the highest incidence in the skin and gastrointestinal tract, followed by ILD, which were affected in 61.1% (95% CI: 47.3–74.1), 44.0% (95% CI: 21.2–68.1) and 16.3% (95% CI: 6.7–28.6) of cases, respectively (Table 3). Nearly all skin and gastrointestinal trAEs were low grade, as grade ≥ 3 events only accounted for 1.7% (95% CI: 0.0–8.9) and 3.6% (95% CI: 0.0–11.5) of cases, respectively, for combination TKI and ICI. The pooled incidences of any-grade skin trAEs (61.1% (95% CI: 47.3–74.1) versus 42.6% (95% CI: 25.5–61.2), OR 1.19, *p* = 0.012) (Appendix A), gastrointestinal trAEs (44% (95% CI: 21.2–68.1) versus 40% (95% CI: 22.5–59.5), OR 1.04, *p* = 0.790) (Appendix A), and ILD (16.3% (95% CI: 6.7–28.6) versus 2.8% (95% CI: 1.5–4.3), OR 1.28, *p* = 0.001) (Appendix A) were in general higher in combination therapy than osimertinib monotherapy, though statistical significance was not observed in gastrointestinal trAEs. Grade ≥ 3 organ-specific trAEs were also all higher in combination therapy than osimertinib monotherapy (skin 1.7% (95% CI: 0.0–8.5) versus 0.2% (95% CI: 0.0–0.9), OR 1.13, *p* = 0.082; gastrointestinal 3.6% (95% CI: 0.0–11.6) versus 1.0% (95% CI: 0.2–2.1), OR 1.13, *p* = 0.076; ILD 4.4% (95% CI: 0.8–9.8) versus 0.5% (95% CI: 0.0–1.5), *p* = 0.003), (Supplementary Appendix A). Furthermore, ILD was the only trAE for sensitivity analysis with both combination and monotherapy groups reported studies with moderate sample size ≥ 40. In such sensitivity analysis, the combination group consistently resulted in a higher risk of ILD of any-grade (OR 1.48 (95% CI: 1.34–1.62), *p* < 0.001) and of grade ≥ 3 (OR 1.24 (95% CI: 1.06–1.45), *p* = 0.007), respectively (Appendix A).

### 3.7. Incidence of trAE Leading to Death and Drug Discontinuation

Only TATTON reported death in a patient after receiving osimertinib plus 3 mg/kg durvalumab due to pneumonia. However, this was not considered related to treatment. No trAE led to death in other combination therapy studies. In trials of osimertinib monotherapy, a pooled proportion of 7% (95% CI: 0–14) of trAEs led to death. It was also revealed that trAE led to discontinuation of ICI in 14% (95% CI: 7–22) of cases and TKI in 12% (95% CI: 2–21) of cases. In osimertinib monotherapy, 7% (95% CI: 0–14) of trAE led to the drug discontinuation.

## 4. Discussion

Immune checkpoint inhibitors, especially anti-PD-(L)1 antibodies, have revolutionized the treatment paradigm of advanced NSCLC, especially those without targetable driver mutations. However, the combinational use of EGFR-TKI and ICI, regardless of whether they are used concomitantly or sequentially, has raised serious concerns over undesirable toxicities. To the best of our knowledge, our study is the first meta-analysis and systematic review of trAEs in combination EGFR-TKI and ICI in advanced EGFR-mutant NSCLC. The most significant finding of the current study was that the pooled incidences of grade ≥ 3 skin and gastrointestinal trAEs and ILD were significantly higher in combination therapy than osimertinib monotherapy. This finding is consistent with previous reports of combining PD-L1 inhibitors and EGFR-TKIs. The recent FAERS database study revealed 25.7% (18 of 70) of patients treated with combination EGFR-TKI and nivolumab experienced ILD of any-grade [36], which echoed our findings (16%, Table 3). Although the incidence of ILD reported here was lower, the actual rate of adverse events may be higher because our analysis considered only the most common organ-specific reported trAE as the overall organ-specific trAE given the unavailability of patients’ individual data from reported studies.

It should be noted that retrospective studies were included in this review to complement both the quantitative meta-analysis and the qualitative discussion of the nature of trAEs. The current evidence of retrospective studies shows a higher risk of ILD and other severe trAEs in the use of osimertinib immediately (within 180 days) after nivolumb [38,39] or other PD-L1 antibodies [37]. The association appeared to be specific for osimertinib, as administration of first or second-generation TKIs immediately after nivolumab did not increase the risk of ILD [39], and no trAEs were observed when PD-L1 blockers were followed by other EGFR-TKIs [37,71]. The reason why ILD development is different between first/second-generation TKI and osimertinib remains unclear. The activation of T-cell effects by ICI may be upregulated to cause ILD in osimertinib but not in first/second-generation TKI [39].

Several case reports also revealed a higher risk of ILD in the sequential use of nivolumab followed by osimertinib within a short interval. The onset of ILD was reported to be on the 63rd day in a 37-day interval [49] and the 50th day in a 46-day interval [50]. Both cases were treated with steroids and ILD remained in remission. It is noteworthy that in a case of osimertinib-induced ILD, osimertinib re-challenge 8 months after the last initiation of nivolumab led to remarkable tumor shrinkage without evidence of relapse of osimertinib-induced ILD [51].

The mechanism of the synergistic toxicity of these two therapies remains undeciphered. It may be related to prolonged receptor occupancy of PD-L1 antibodies [37,50], higher levels of proinflammatory cytokines observed in the combination therapy [43], and enhanced T-cell mediated killing of tumor cells by cytokine-induced MHC class-I presentation [42]. Nivolumab can occupy PD-1 receptors on T-cells for approximately two months, which may contribute to delayed trAEs. A recent study of liquid chromatography-mass spectrometry analyses revealed that nivolumab concentrations lower than the trough concentrations during nivolumab and osimertinib treatment were high enough to induce ILD [50].

Our study also revealed increased risks of high-grade skin and gastrointestinal AEs in combination TKI and ICI versus TKI alone, albeit a weak association. Other organ-specific toxicities such as hepatotoxicity were not included into this meta-analysis due to limited data available. Increased levels of alanine aminotransferase up to 40% were reported in patients receiving a combination of gefitinib and durvalumab [72].

Our analysis has some limitations. First, our findings mainly focused on trials of combination osimertinib plus durvalumab. Additional data will be needed to clarify the relative risks of other TKIs following PD-L1 blockade. The exclusion of case reports may result in selection bias, even though the ability to draw statistically meaningful conclusions from case reports is extremely limited. Further, there were only a few randomized and prospective clinical trials studying the use of ICI and EGFR-TKI for advanced NSCLC. Thus, the number of studies included in this meta-analysis is small. The limited patient size also hinders us from performing clinically meaningful statistical analysis to further compare the risk of AEs between concurrent and sequential treatment of the two agents. In the sensitivity analysis which considered studies with moderate sample size ≥ 40, the conclusion remained that combination therapy generated higher risk of ILD when compared with monotherapy. While ILD was reported to be higher in the Japanese population [34], subgroup analysis was not possible due to the lack of reporting on patient demographics. In addition, although ILD has been reported to be more frequent in patients treated with osimertinib, it is difficult to examine the difference in the incidence of ILD among every TKIs. While most studies that reported ILD involved combination ICI and osimertinib [34,35,38], only one investigated erlotinib [70] while the others did not state the TKI that caused ILD [36,37,39,71]. Larger prospective studies are warranted to investigate the incidence of trAEs between concurrent and sequential treatment of ICI and TKI, and to stratify the risks of ILD according to different ethnicities and TKIs. Furthermore, we were unable to investigate and compare the incidences of other clinically important trAEs including endocrinopathies, hepatotoxicities, nephrotoxicities, and neurotoxicities since they were rarely and less systematically reported in every selected study. Moreover, toxicities in some patients might have been attributed to other causes such as progressive disease of their underlying NSCLC, which would underestimate the reported treatment-related toxicity and were inherent limitations of retrospective analyses. Lastly, our study results cannot serve as a recommendation guideline on how and when to administer TKIs and ICI judiciously, as these agents could be safely administered in some patients in the clinical trials included in this meta-analysis.

## 5. Conclusions

Our meta-analysis demonstrates, in general, an increased overall incidence of trAE and organ-specific trAE in combination therapy with EGFR-TKI and ICI. Future prospective RCTs are warranted to determine the most optimal drug of choice of ICI and the best timing of initiation of ICI after failure to prior EGFR-TKI.

## Figures and Tables

**Figure 1 cancers-14-02157-f001:**
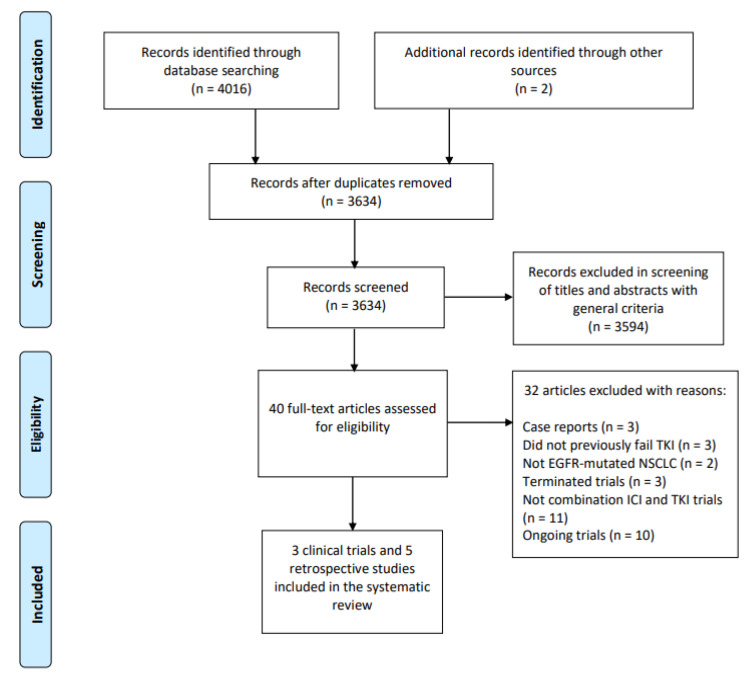
PRISMA flow chart showing the identification and selection of included studies.

**Table 1 cancers-14-02157-t001:** Summary of results from included clinical trials of epidermal growth receptor tyrosine kinase inhibitors in combination with immune checkpoint inhibitors.

Clinical Trial	Author	Phase	Key Eligibility Criteria	Treatment Arms	Primary Objective	Treatment Line	Median Age (Range), Years)	Sample Size (Female%)	Grade 3/4 trAE	Discontinuation Due to Adverse Events	ORR	PFS (Months)	DOR (Months)
TATTON/NCT02143466	Oxnard et al.	Ib	Advanced EGFR-mutant NSCLC and disease progression to a prior TKI	Arm 1: osimertinib 80 mg daily + durvalumab 3 mg/kg every 2 weeks (*n* = 10) Arm 2: osimertinib 80 mg daily + durvalumab 10 mg/kg every 2 weeks (dose escalation)	Safety, tolerability	≥s	Arm 1: 67 (46–78) Arm 2: 58 (44–73)	Arm 1: 10 (70%) Arm 2: 13 (54%)	Arm 1: 60%, Arm 2: 38.5%	Arm 1: 30% ^a^, 40% ^b^ Arm 2: 23.1% ^a^, 38.5% ^b^	Arm 1: 40%, Arm 2: 39%	NA	NA
CheckMate012/NCT01454102	Gettinger et al.	I	EGFR-mutant chemotherapy-naïve, EGFR-TKI naïve or TKI treated stage IIIB/IV NSCLC	Nivolumab 3 mg/kg every 2 weeks and erlotinib 150 mg/d	Safety, tolerability	≥s	63 (41–80)	21 (62%)	24%	10%	15%	48	5.1 ^c^, 5.7 ^d^
CAURAL/NCT02454933	Yang et al.	III	EGFR T790M-positive, TKI-treated NSCLC	Arm 1: Osimertinib (80 mg once daily) Arm 2: osimertinib (80 mg once daily) + durvalumab (10 mg/kg IV every 2 weeks)	Safety, tolerability	≥s	Arm 1: 65 (41–80) Arm2: 56 (41–78)	Arm 1: 17 (76%) Arm 2: 12 (50%)	34%	17%	80% ^a^, 64% ^e^	NA	NA

DOR, duration of response; EGFR, epidermal growth factor receptor; NA, not available; NSCLC, non-small cell lung cancer; PFS, progression-free survival; ORR, objective response rate; trAE, treatment-related adverse events; TKI, tyrosine kinase inhibitor. ^a^ Attributed to osimertinib. ^b^ Attributed to durvalumab. ^c^ Attributed to nivolumab. ^d^ Attributed to erlotinib. ^e^ Attributed to osimertinib in combination with durvalumab.

**Table 2 cancers-14-02157-t002:** Summary of included retrospective studies.

Author	Year	Title	Study Design	Inclusion Criteria	EGFR Mutant (%)	Treatment Line	Treatment Arms	Time between ICI and TKI (Days)	Median Age (Range), Years	Sample Size (Female %)	Adverse Event Reported
Ito et al.	2022	Treatment with immune checkpoint inhibitors after EGFR-TKIs in EGFR-mutated lung cancer	Multiple institutions (Iwate Medical University Hospital, Iwate Prefectural Central Hospital, and Miyagi Cancer Center) retrospective study	EGFR mutant NSCLC previously treated with TKI	Ex19del/ L858R (25%) T790M (8%)	≥s	G/E/Af/O followed by N/P/At	139 (1–707)	67 (38–80)	25 (40%)	ILD
Schoenfeld et al.	2019	Severe immune-related adverse events are common with sequential PD-(L)1 blockade and osimertinib	Single institution (Memorial Sloan Kettering Cancer Center) retrospective study	EGFR mutant NSCLC treated with PD-(L)1 blockers and EGFR-TKI irrespective of drug sequence	NA	s	Arm 1: N/P/At/D followed by O Arm 2: sequential osimertinib followed by N/P/At/D	Arm 1: 61 (12–1446) Arm 2: 5 (1–256)	Arm 1: 61 (30–79) Arm 2: 56 (36–85)	Arm 1: 41 (66%)Arm 2: 29 (83%	Severe immune related adverse events (4 G3 pneumonitis, 1 G3 colitis, 1 G4 hepatitis)
Uchida et al.	2019	Different incidence of interstitial lung disease according to different kinds of EGFR-tyrosine kinase inhibitors administered immediately before and/or after anti-PD-1 antibodies in lung cancer	Single institution (Saitama medical university international medical center) retrospective study	Advanced EGFR mutant patients who receive TKI immediately before and/or after N or P	Ex19del/ L858R (100%)	≥s	O or Af before or after N or P	NA	69 (44–80)	26 (62%)	ILD
Oshima et al.	2018	EGFR–TKI-associated interstitial pneumonitis in nivolumab-treated patients with non–small cell lung cancer	Database study of US FDA Adverse Event Reporting System	EGFR mutant NSCLC	NA	NA	N/P/At in combination with Af/E/G/O	NA	Without N and TKI: 63 (NA) Without N, with TKI: 69 (NA). With N, without TKI: 66 (NA) With N and TKI: 64 (NA)	20516	ILD
Kotake et al.	2017	High incidence of interstitial lung disease following practical use of osimertinib in patients who had undergone immediate prior nivolumab therapy	Single institution (Shizuoka Cancer center) retrospective study	Advanced EGFR mutant NSCLC and disease progression on or after EGFR TKI	T790M (100%)	≥s	N followed by O	In patients with ILD: 14 (7–28) In patients without ILD: 49 (28–119)	Median age NA In patients with ILD: 4 < 70 In patients without ILD: 10 < 70, 5 > 70	19	ILD

Af, afatinib; At, atezolizumab; D, durvalumab; E, erlotinib; EGFR, epidermal growth factor receptor; G, gefitinib; G, grade; ICI, immune checkpoint inhibitor; ILD, interstial lung disease; NSCLC, non-small cell lung cancer; NA, not availabe; N, nivolumab; O, osimertinib; P, pembrolizumab; TKI, tyrosine kinase inhibitor.

**Table 3 cancers-14-02157-t003:** Pooled incidence of trAE in combination epidermal growth receptor tyrosine kinase inhibitors and immune checkpoint inhibitors versus tyrosine kinase inhibitor monotherapy.

trAEs	Combination of TKI and ICI	TKI Monotherapy	Odds Ratio (Combined vs. Monotherapy)	*p*
Overall				
Any grade	100.0 (96.3, 100.0) ^a,b^	87.7 (68.1, 99.0) ^b,c^	1.27 (0.75, 1.66)	0.077
Grade ≥ 3	30.0 (12.0, 51.6) ^b,c^	13.8 (0.1, 40.4) ^b,c^	1.23 (0.85, 1.76)	0.271
Skin				
Any grade	61.1 (47.3, 74.1) ^a^	42.6 (25.5, 61.2) ^c^	1.19 (0.95, 1.49)	0.012
Grade ≥ 3	1.7 (0.0, 8.5) ^a^	0.2 (0.0, 0.9) ^a^	1.13 (0.96, 1.29)	0.082
Gastrointestinal				
Any grade	44.0 (21.2, 68.1) ^c^	40.3 (22.5, 59.5) ^c^	1.04 (0.77, 1.40)	0.790
Grade ≥ 3	3.6 (0.0, 11.6) ^a^	1.0 (0.2, 2.1) ^a^	1.13 (0.99, 1.02)	0.076
Interstitial lung disease (ILD)				
Any grade	16.3 (6.7, 28.6) ^a^	2.8 (1.5, 4.3) ^a^	1.28 (1.11, 1.48)	0.001
Grade ≥ 3	4.4 (0.8, 9.8) ^a^	0.5 (0.0, 1.5) ^a^	1.16 (1.05, 1.28)	0.003
Sensitivity analysis (on studies of sample size >40)				
Any grade	30.5 (23.1, 38.3) ^a^	3.4 (2.0, 5.0) ^a^	1.48 (1.34, 1.62)	< 0.001
Grade ≥ 3	9.6 (2.7, 23.1) ^a^	1.0 (0.3, 2.1) ^a^	1.24 (1.06, 1.45)	0.007

ICI, immune checkpoint inhibitor; TKI, tyrosine kinase inhibitor; trAE, treatment-related adverse events. ^a^ Estimated from fixed effect model. ^b^ Results from Nie et al. [47] were not included into calculation as overall trAEs were not reported. ^c^ Estimated from random effects model due to heterogeneity.

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
