# Peer review of "Treatment-Related Adverse Events of Combination EGFR Tyrosine Kinase Inhibitor and Immune Checkpoint Inhibitor in EGFR-Mutant Advanced Non-Small Cell Lung Cancer: A Systematic Review and Meta-Analysis"

_cancers, 2022, doi:10.3390/cancers14092157_

Round 1

Reviewer 1 Report

The authors precisely replied to my comments. Revised manuscript is suitable for publication.

Reviewer 2 Report

All I pointed out in the last review are corrected or responded briefly.

This manuscript is a resubmission of an earlier submission. The following is a list of the peer review reports and author responses from that submission.

Round 1

Reviewer 1 Report

Dear authors, it was a pleasure to review this manuscript.

I have some concerns:

  • In Figure 1, the numbers do not add up, please review them.
  • The authors show the characteristics of the trial where Osimertinib is used as monotherapy in Supp Table1. For the sake of clarity, they should show the characteristics of the patients treatment in the combination trials too. 
  • In the Results 3.1 paragraph, the authors explained the exclusion criteria for 16 data sources (of the 3621 record excluded). It would be better to clarify these exclusion criteria, even just by dividing them by groups and adding them in the Figure 1 PRISMA flow chart. E.g. Record Excluded 3622 (or 3621?): - Ongoing trials (11) etc
  • About ongoing trials, a supplementary table with the list and some basic characteristics would be very much appreciated
  • The authors should add, if the data is available, the wash out time between the TKI and the ICI start in the sequential treatment 
  • In the paragraph where incidence of organ-specific trAEs are discussed, the authors declare that the highest incidence in the TKI and ICI combination group is in the skin and gastrointestinal tract. Is this result due to lack of data about, for example, endocrinopathies?  If yes, it would be better to explain clearly to the readers that skin and GI toxicities are the most frequent, taking into account that endocrinopathies, neurological toxicities, etc were not addressed for lack of data
  • In the discussion, the authors may want to add some data from Lizotte et al. Cancer Immunol Res 2018, and Uchida et al 2019 (already present in the ref) where a possible mechanism to explain the increased toxicities is described
  • Finally, can the author confirm the data included in Suppl Table 4? No star(s) for Outcome data etc?
  • There are some English mistakes and misspelling/mistyping in the manuscript, please review them 

Reviewer 2 Report

Summary

This study compares proportions of trAEs between TKI +/- ICI in EGFR-mutant NSCLC patients from 3 (phase IB-III) clinical trials and 5 retrospective studies with median FU of 19 months. It is an important topic that will be of interest to any oncologist involved with treatment of lung cancers. The authors say this is the first meta-analysis comparing TKI +/- ICI in EGFR-mutant NSCLC. They acknowledge the extreme heterogeneity of the studies considered. Case reports were excluded. 

General Comments

Overall the manuscript is well written. The studies included for analysis seem complete. The only study I am aware of that does not seem to be included is Lisberg et al (PMID 29874546) so I would be interested in seeing if it was excluded or was not captured in the search.

Specific Comments

Line 115 - was stage IIIC included?
Table 2 - sample size is different from the cited paper.
Table 3 - there are some rounding errors in this table. E.g. for Grade >= 3 combination TKI and ICI from 30.15 is rounded to 30.0. Please check for the rest.

Reviewer 3 Report

Chan D WK, et al. have reported a title of “Treatment-related Adverse Events of Combination EGFR Tyrosine Kinase Inhibitor and Immune Checkpoint Inhibitor in EGFR-mutant Advanced Non-small Cell Lung Cancer: A Systematic Review and Meta-Analysis”.

  This report contains clinical meaningful findings . However, there are serious concerns in this manuscript.

  1. This meta-analysis contains small number of studies. The results and conclusions in the present analysis are not robust.

  1. Previous studies have suggested that incidence of interstitial lung disease (ILD) in patients treated with osimertinib may be higher than that in patients treated with other EGFR-TKIs. The authors should examine the difference in incidence of ILD according to EGFR-TKIs.

  1. Ito’s study included patients treated with ICIs after EGFR-TKIs. This study should be excluded from the present meta-analysis.

Reviewer 4 Report

Authors demonstrated the combination of EGFR-TKI and ICI would have higher risk of treatment-related adverse events than EGFR-TKI mono-therapy. the theme of this manuscript seems interesting. I would like to make some comments on it.

First, No description is provided in the manuscript regarding how the data for the TKI mono-therapy group was selected. The FLAURA study data suddenly appears, but it is not clearly stated on what basis this study data was selected and other study data was not selected. Moreover, any grade of trAEs are discussed in the results, but the most clinically important side effects are those that are Grade 3 or higher. Figures of Forest-plots should be changed the data to the Grade ≥3 cases. Besides, although not discussed in this manuscript regarding dosage, it appears that higher durvalumab doses are associated with more frequent side effects according to TATTON 2019. If there is a correlation between dosage and side effects, dosage in other papers should be discussed as well. Moreover, based on this manuscript, it seems that the frequency of ILD is considerably higher in the concurrent group than in the TKI alone group, but the risk of concurrent administration cannot be evaluated because the frequency of occurrence by ICI was not evaluated.

Minor point: In table 2, the sample size of "Oshima et al" paper is written as 29516, but it should be 20516.